# Estimation of Two Component Activities of Binary Liquid Alloys by the Pair Potential Energy Containing a Polynomial of the Partial Radial Distribution Function

**Jiulong Hang and Dongping Tao \***

Faculty of Metallurgical and Energy Engineering, Kunming University of Science and Technology, Kunming 650093, China

\* Correspondence: dongpingt@aliyun.com

**Abstract:** An investigation of partial radial distribution functions and atomic pair potentials within a system has established that the existing potential functions are rooted in the assumption of a static arrangement of atoms, overlooking their distribution and vibration. In this study, Hill's proposed radial distribution function polynomials are applied for the pure gaseous state to a binary liquid alloy to derive the pair potential energy. The partial radial distribution functions of 36 binary liquid alloy from literatures were used to obtain the binary model parameters of four thermodynamic models for validation. Results show that the regular solution model (RSM) and molecular interaction volume model (MIVM) outperform other models when the asymmetric method calculates the partial radial distribution function. RSM demonstrates an average SD of 0.078 and an ARD of 32.2%. Similarly, MIVM exhibits an average SD of 0.095 and an average ARD of 32.2%. Wilson model yields an average SD of 0.124 and an average ARD of 226%. Nonrandom two-liquid (NRTL) model exhibits an average SD of 0.225 and an average ARD of 911%. On applying the partial radial distribution function symmetry method, MIVM and RSM outperform the other models, with an average SD of 0.143 and an average ARD of 165.9% for MIVM. RSM yields an average SD of 0.117 and an average ARD of 208.3%. Wilson model exhibits average values of 0.133 and 305.6% for SD and ARD, respectively. NRTL model shows an average SD of 0.200 and an average ARD of 771.8%. Based on this result, the influence of the symmetry degree on the thermodynamic model is explored by examining the symmetry degree as defined by the experimental activity curves of the two components.

**Keywords:** pair potential; partial radial distribution function; binary liquid alloy; activity

## 1. Introduction

Pair potentials play a fundamental role in comprehending the static and dynamic properties of gases, liquids, and solids [1]. The main objective of studying pair potentials is to develop accurate and reliable models that describe interatomic interactions and enable simulation, prediction, and explanation of the behavior and properties of atomic systems. Various computational simulation studies, such as for protein folding, drug–receptor interactions, and material property prediction, are conducted using atomic pair potentials to ascertain the underlying mechanisms of atomic interactions, explore novel material properties, and optimize drug design. Materials science and surface science researchers have investigated atomic potentials to thoroughly understand phenomena such as material structure, stability, crystal defects, and surface adsorption. Understanding interatomic interactions is essential in material design, catalyst development, and comprehending material interfaces and surface phenomena. Atomic pair potential models encompass widely used semiempirical potentials such as the Lennard–Jones potential [2], Morse potential [3], and Born–Mayer potential [4], as well as atomic force fields, quantum mechanical descriptions, and statistical mechanics methods. Different factors affect the selection and examination of these models, including application needs and accuracy considerations [5–7]. Accurate pair

potentials provide a comprehensive description of energy, geometric structure, and spectral properties of molecules and serve as the basis for investigating collision and chemical-reaction dynamics, such as those for atomic collisions. Thus, an in-depth study of potential pair functions holds significant physical implications and offers broad applications. This study reveals a direct correlation between the radial distribution function (RDF) and atomic pair potential. This paper proposes a linkage between the radial distribution function and the atomic pair potential energy. This is applied to a binary liquid alloy system using a polynomial expression for the radial distribution function of a pure gas. The group element activity is estimated based on the symmetries outlined in this paper, utilizing the radial distribution functions of 36 binary liquid alloys from the literature into the model.

## 2. Thermodynamic Model and Symmetry

### 2.1. Miedema Model

In 1973, Miedema et al. suggested a semi-empirical theoretical model that effectively characterizes the heat of mixing in binary alloys. This model encompasses solid solubility, the mesostable positioning of metal ions following ion injection, surface aggregation phenomena, surface energy, the formation energy of single vacancies in metals and metal compounds, as well as the development of nonphenolic alloys. For any metal binary alloy, the expression is [8]

$$f_{ij} = \frac{2pV_{mi}^{2/3}V_{mj}^{2/3} \times [(q/p)(\Delta n_{ws}^{1/3})^2 - (\Delta \varphi)^2 - a(r/p)]}{(\Delta n_{ws}^{1/3})_i^{-1} + (\Delta n_{ws}^{1/3})_j^{-1}} \tag{1}$$

$$\Delta H_m = f_{ij} \frac{x_i[1 + \mu_i x_j(\varphi_i - \varphi_j)]x_j[1 + \mu_j x_i(\varphi_j - \varphi_i)]}{x_i V_{mi}^{2/3}[1 + \mu_i x_j(\varphi_i - \varphi_j)] + x_j V_{mj}^{2/3}[1 + \mu_j x_i(\varphi_j - \varphi_i)]} \tag{2}$$

The activity coefficient is:

$$\ln \gamma_i = \frac{\Delta H_m}{RT} \left\{ 1 + (1 - x_i) \left\{ \begin{array}{c} \dfrac{1}{x_i} - \dfrac{1}{1 - x_i} - \dfrac{\mu_i(\varphi_i - \varphi_j)}{1 + \mu_i(1 - x_i)(\varphi_i - \varphi_j)} + \dfrac{\mu_j(\varphi_j - \varphi_i)}{1 + \mu_j x_i(\varphi_j - \varphi_i)} \\[2mm] - \dfrac{V_{mi}^{2/3}[1 + \mu_i(1 - 2x_i)(\varphi_i - \varphi_j)] + V_{mj}^{2/3}[-1 + \mu_j(1 - 2x_i)(\varphi_j - \varphi_i)]}{x_i V_{mi}^{2/3}[1 + \mu_i x_j(\varphi_i - \varphi_j)] + x_j V_{mj}^{2/3}[1 + \mu_j x_i(\varphi_j - \varphi_i)]} \end{array} \right\} \right\} \tag{3}$$

In the above equation, $x_i$ and $x_j$ represent the mole fractions of species $i$ and $j$ respectively. $V_{mi}$ and $V_{mj}$ denote the molar volumes of $i$ and $j$, while $(n_{ws})_i$ and $(n_{ws})_j$ stand for the electron densities, $\varphi_i$ and $\varphi_j$ are electronegative. The variables $p,q,r,\mu_i,\mu_j$ and $a$ are empirical constants. For a binary alloy composed of a transition group metal and a multivalent non-transition group metal, the formula $f_{ij}$ is transformed to:

$$f_{ij} = \frac{2pV_{mi}^{2/3}V_{mj}^{2/3} \times [(q/p)(\Delta n_{ws}^{1/3})^2 - (\Delta \varphi)^2 - \alpha(r/p)_i(r/p)_j]}{(\Delta n_{ws}^{1/3})_i^{-1} + (\Delta n_{ws}^{1/3})_j^{-1}} \tag{4}$$

### 2.2. Regular Solution Model (RSM)

Hildebrand suggested the RSM in 1929 [9]. According to this model, the molar excess volume and mixing entropy are zero, and the molar excess mixing Gibbs free energy is equal to the molar excess mixing enthalpy [10]. The expression for the molar excess Gibbs free energy for a binary system is as follows; the activity coefficient of the component $i$ is $\ln \gamma_i$.

$$\frac{G_m^E}{RT} = \Delta H_m = \Omega_{ij} x_i x_j \tag{5}$$

$$\ln \gamma_i = \Omega_{ij} x_j^2 \tag{6}$$

Here, $x_i$ and $x_i$ are the mole fractions of components $i$ and $j$, respectively, and $\Omega_{ij}$ is the interaction parameter between components $i$ and $j$. $\Omega_{ij}$ is only related to temperature and does not change with the composition of components.

### 2.3. Wilson Model

In 1964, Wilson [11] suggested a semiempirical and semitheoretical model based on the local concept. This model assumes that "local concentrations" (represented as volume fractions) primarily determine molecular interactions. These concentrations are defined in relation to the Boltzmann distribution probability term for energy. The excess free energy associated with the concentrations can be expressed as follows.

$$\frac{G_m^E}{RT} = -x_i \ln(x_i + A_{ji}x_j) - x_j \ln(x_j + A_{ij}x_i) \tag{7}$$

$$\ln \gamma_i = -\ln(x_i + A_{ij}x_j) + x_j \left( \frac{A_{ij}}{x_i + A_{ij}x_j} - \frac{A_{ji}}{x_j + A_{ji}x_i} \right) \tag{8}$$

Here, $A_{ij}$ and $A_{ji}$ are the interaction parameters between components $i$ and $j$, which are only related to temperature and do not change with the composition of components [12].

### 2.4. Nonrandom Two-Liquid (NRTL) Model

The NRTL model, suggested by Renon and Prausnitz in 1968 [13], has been extensively used in correlating thermodynamic data, computing thermodynamic properties, and predicting phase equilibrium for diverse fluid systems in chemical processes. This model, based on the concept of local concentration, permits the determination of molar excess Gibbs free energy. The activity coefficient of the component $i$ is $\ln \gamma_i$.

$$\frac{G_m^E}{RT} = x_i x_j \left( \frac{\tau_{ji} G_{ji}}{x_i + x_j G_{ji}} + \frac{\tau_{ij} G_{ij}}{x_j + x_i G_{ij}} \right) \tag{9}$$

$$\ln \gamma_i = x_j^2 \left[ \frac{\tau_{ji} G_{ji}^2}{(x_i + x_j G_{ji})^2} + \frac{\tau_{ij} G_{ij}}{(x_j + x_i G_{ij})^2} \right] \tag{10}$$

$$\tau_{ij} = \frac{g_{ij} - g_{jj}}{RT} \quad \tau_{ji} = \frac{g_{ji} - g_{ii}}{RT} \tag{11}$$

$$G_{ij} = \exp(-\alpha \tau_{ij}) \quad G_{ji} = \exp(-\alpha \tau_{ji}) \tag{12}$$

Here, $G_{ij}$ and $G_{ji}$ are energy parameters characterizing the interaction between components $i$ and $j$; $\alpha$ is related to nonrandomness in the mixture, independent of temperature and composition of a solution. Moreover, the characteristic parameter of a solution depends on the solution type. When the mixture is entirely random, many binary system experimental data show that $\alpha$ varies from 0.2 to 0.47. In this study, $\alpha = 0.3$.

### 2.5. Molecular Interaction Volume Model (MIVM)

In 2000, Tao suggested the molecular interaction volume model [14], which applied statistical thermodynamics and fluid phase equilibrium theory to describe the motion of liquid molecules. This model yields the following expression:

$$\frac{G_m^E}{RT} = x_i \ln \left( \frac{V_{mi}}{x_i V_{mi} + x_j V_{mj} B_{ji}} \right) + x_j \ln \left( \frac{V_{mj}}{x_j V_{mj} + x_i V_{mi} B_{ij}} \right) - \frac{x_i x_j}{2} \left( \frac{Z_i B_{ji} \ln B_{ji}}{x_i + x_j B_{ji}} + \frac{Z_j B_{ij} \ln B_{ij}}{x_j + x_i B_{ij}} \right) \tag{13}$$

The activity coefficients of the components $i$ is $\ln \gamma_i$.

$$\ln \gamma_i = 1 + \ln \left( \frac{V_{mi}}{V_{mi}x_i + V_{mj}B_{ji}x_j} \right) - \frac{x_i V_{mi}}{V_{mi}x_i + x_j V_{mj}B_{ji}} - \frac{x_j V_{mi} B_{ij}}{V_{mj}x_j + x_i V_{mi} B_{ij}}$$

$$- \frac{x_j^2}{2} \left[ \frac{Z_i B_{ji}^2 \ln B_{ji}}{(x_i + x_j B_{ji})^2} + \frac{Z_j B_{ij} \ln B_{ij}}{(x_j + B_{ij}x_i)^2} \right] \quad (14)$$

Here, $Z_i$ and $Z_j$ are the first coordination numbers of $i$ and $j$ pure substances, respectively, and $V_{mi}$ and $V_{mj}$ are the molar volumes of $i$ and $j$, respectively. $B_{ij}$ and $B_{ji}$ are the interaction parameters of $i - j$ and $j - i$, respectively. $k$ is the Boltzmann constant, and $T$ is the temperature.

*2.6. Symmetry*

Figure 1 shows the molar fraction $x_i = x_j = 0.5$ of the two components as the symmetry axis of their activity curve. The symmetry degree of the activity curve can be defined as the average absolute value of the activity difference between the two components at the same concentration. In this context, $S$ represents the measure of symmetry.

$$S_{ij} = \frac{\sum\limits_{l=1}^{m} \left| (a_i - a_j)_{x_i=x_j} \right|_l}{m} \quad (15)$$

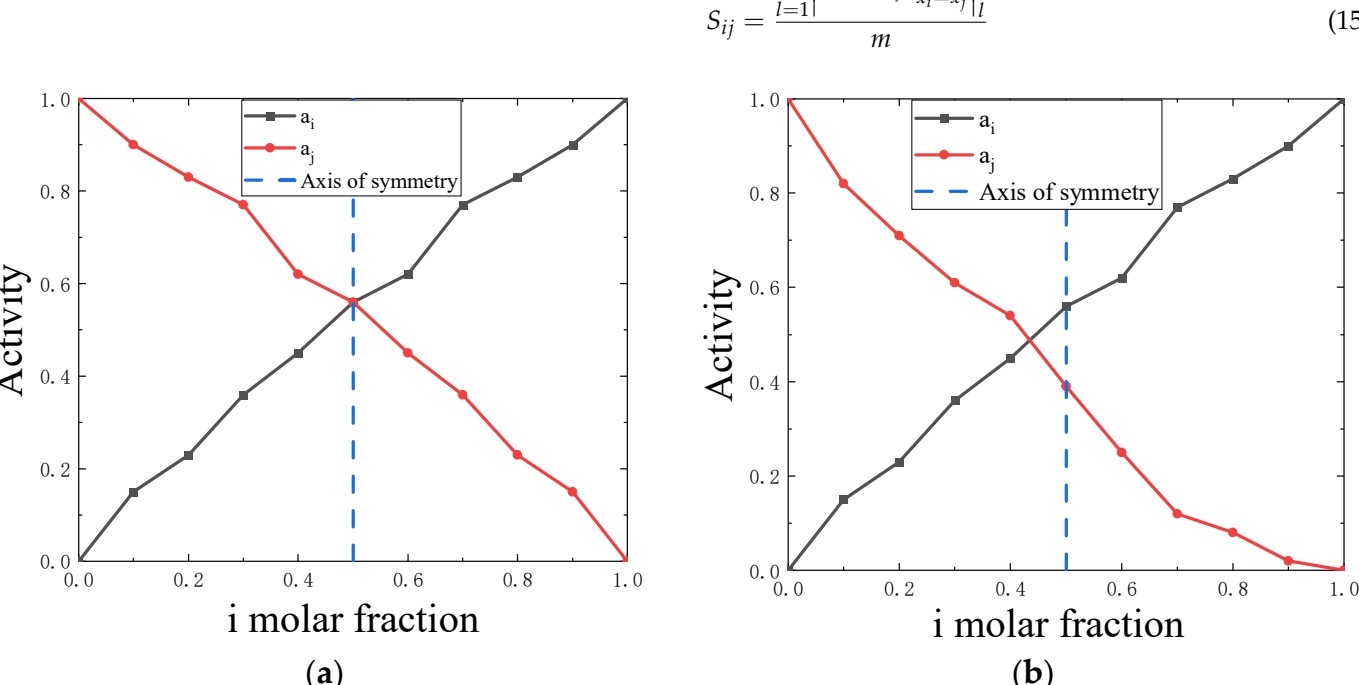

**Figure 1.** (**a**) Schematic diagram of symmetrical activity curves, (**b**) Schematic diagram of asymmetric activity curves.

The symmetry increases as the $S$ value decreases. At $S = 0$, the binary liquid alloy exhibits complete symmetry, where $a_i$ and $a_j$ denote the experimental activity of components $i$ and $j$, respectively, and $m$ represents the number of experimental activities. This definition also applies to similar geometric figures. The binary liquid alloy with low symmetry is shown in Figure 2. Table 1 presents the symmetry degrees of the activity curves for the 36 binary liquid alloys based on the definitions mentioned above.

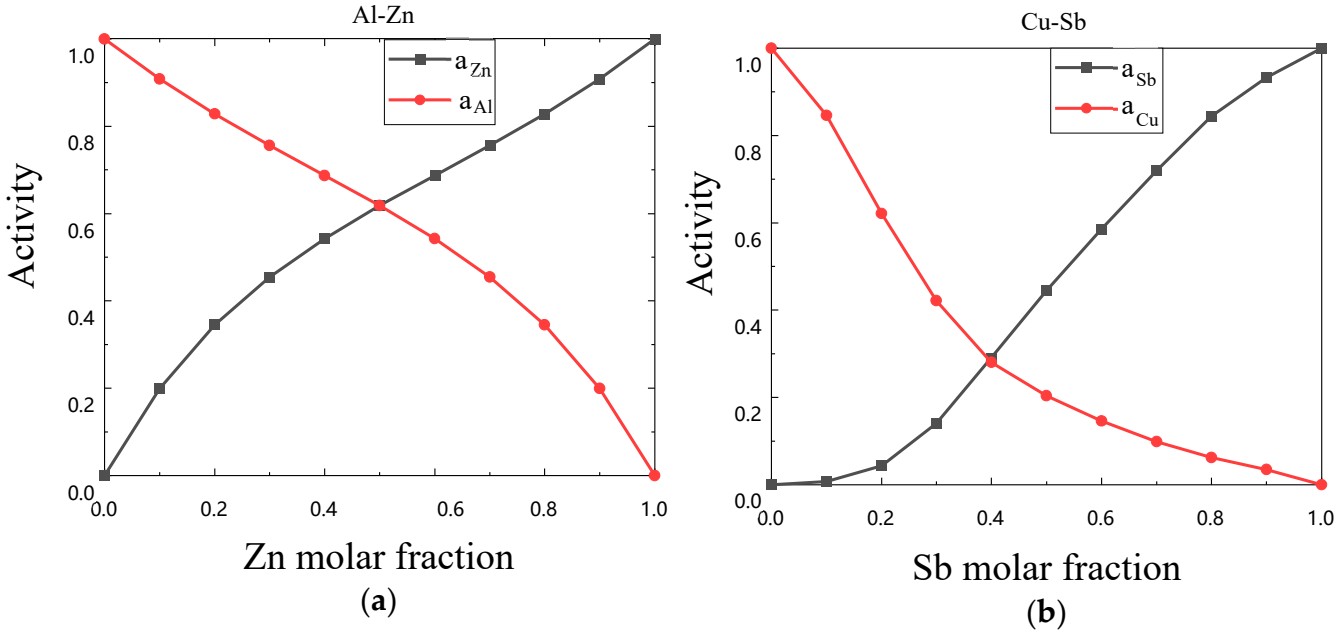

**Figure 2.** (**a**) Schematic diagram of symmetrical Al-Zn activity curves, (**b**) Schematic diagram of asymmetrical Cu-Sb activity curves.

**Table 1.** Symmetry of the 36 binary liquid alloys from highest to lowest.

| System | Co-Ni | Al-Zn | Cu-Ni | Al-Ni | Cu-Fe | Ge-Sn | Ag-Cu | Pb-Sb | Al-Si | Al-Co |
|---|---|---|---|---|---|---|---|---|---|---|
| $S_{ij}$ | 0 | 0 | 0.0028 | 0.0034 | 0.0048 | 0.007 | 0.0088 | 0.0096 | 0.0102 | 0.0114 |
| System | Li-Mg | Sb-Sn | Cu-Zr | K-Na | Pb-Sn | Al-Mg | Cs-K | Li-Na | Au-Cu | Al-Li |
| $S_{ij}$ | 0.0114 | 0.0115 | 0.0119 | 0.0136 | 0.0158 | 0.0177 | 0.0228 | 0.0336 | 0.0364 | 0.0452 |
| System | Nb-Zr | Ni-Pd | Cu-Mg | Al-Ca | Ni-Zr | Al-Sn | Al-Cu | Nb-Ni | Cu-Sn | Au-Si |
| $S_{ij}$ | 0.0473 | 0.0494 | 0.0597 | 0.0724 | 0.0807 | 0.0823 | 0.1116 | 0.1334 | 0.1342 | 0.139 |
| System | Li-Sn | Fe-Si | Ag-In | Ge-Te | Al-Au | Cu-Sb | | | | |
| $S_{ij}$ | 0.14 | 0.1454 | 0.1483 | 0.1548 | 0.160 | 0.208 | | | | |

## 3. Pair Potential Energy Polynomials of the Binary Liquid

In an extremely diluted pure gas, the correlation between the interatomic potential energy and RDF can be expressed as follows [15]:

$$\lim_{\rho_0 \to 0} g_{12}(r) = \exp\left(-\frac{\varepsilon_{12}}{kT}\right) \tag{16}$$

However, in practical scenarios, the RDF and interatomic potential energy strongly depend on the number density of the system. The relation between the RDF and pair potential energy can be expanded using a polynomial expression in terms of the number density $\rho_0$:

$$g_{12}(r, \rho_0, T) = e^{-\varepsilon/kT}\left[1 + \rho_0 g_1(R, T) + \rho_0^2 g_2(R, T) + \cdots\right] \tag{17}$$

It is seen formula Equation (17), that there is connection between the RDF and pair potential energy. In the case of $\rho_0 \neq 0$, 3 atoms in the pure gas system, labeled 1, 2 and 3, are shown in Figure 3a. These three atoms have six interactions: 1-1, 1-2, 1-3, 2-2, 2-3, 3-3, which influence each other. The radial distribution function for 1-2 is $g_{12}(r, \rho_0, T)$, i.e., the distribution of atom 1 centred on atom 2, but influenced by atom 3. This form is centred on atom 3 and influences atom 1 and atom 2 in $g_{12}(r, \rho_0, T)$.

$$g_{12}(r, \rho_0, T) = e^{-\varepsilon_{12}/kT} \exp\left\{\rho_0 \int_V [e^{-\varepsilon_{13}/kT} - 1][e^{-\varepsilon_{23}/kT} - 1]dr_3\right\}$$
$$= e^{-\varepsilon_{12}/kT}\left\{1 + \rho_0 \int_V [e^{-\varepsilon_{13}/kT} - 1][e^{-\varepsilon_{23}/kT} - 1]dr_3\right\} \tag{18}$$

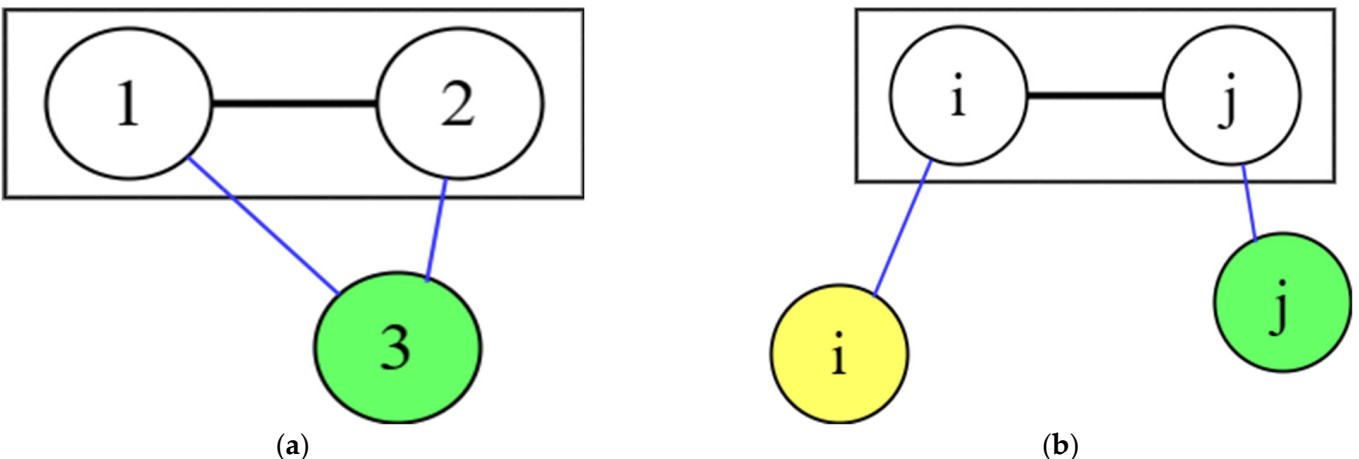

(a)　　　　　　　　　　　　　　　　　　(b)

**Figure 3.** (**a**) Schematic diagram of pure gases or pure liquid atomics interactions, (**b**) Schematic diagram of binary liquid alloy atomics interactions.

Consequently, the expansion based on $\rho_0$ is $1 + \rho_0 \int_V [e^{-\varepsilon_{13}/kT} - 1][e^{-\varepsilon_{23}/kT} - 1]dr_3$. Equation (18) is compared with Equation (17).

$$g_1(r_{12}, T) = \int_V [e^{-\varepsilon_{13}/kT} - 1][e^{-\varepsilon_{23}/kT} - 1]dr_3 = \int_V [e^{-\varepsilon_{13}/kT}e^{-\varepsilon_{23}/kT} - e^{-\varepsilon_{23}/kT} - e^{-\varepsilon_{13}/kT} + 1]dr_3$$
$$= \int_V e^{-\varepsilon_{13}/kT}e^{-\varepsilon_{23}/kT}dr_3 - \int_V e^{-\varepsilon_{23}/kT}dr_3 - \int_V e^{-\varepsilon_{13}/kT}dr_3 + \int_V dr_3 \tag{19}$$

The above is the influence of atomic interactions in pure gases; assume let the interaction of atoms is applied to the liquid metal system, and Equations (16) and (17) are applied to the liquid metal. The interactions between atoms in a pure liquid metal are similar to those in the pure gaseous state, both being interactions between the same atoms. However, there are differences in the binary liquid alloy and different interactions between different atoms, as shown in Figure 3b. There are three atomic interactions in binary liquid alloys: $i - i$, $i - j$, and $j - j$. The RDF $i - j$ describes atomic distribution $i$ around atom $j$ in a binary system. However, the remaining atom $i$ influences the atom $i$ in the RDF $i - j$, while the remaining atom $j$ influences the atom $j$ in the RDF $i - j$.

Unlike pure gas centered on the remaining atoms 3 affecting the RDF 1–2, as shown in Figure 3a. In binary liquid alloys, the remaining atoms i and j influence the atoms i and j in the radial distribution function $i - j$. That is, the atom $i$ in the RDF influencing $i - j$ is centered on the other atoms $i$, while the atom $j$ in the RDF influencing $i - j$ is cetered on the other atoms $j$.

If the RDF in a binary liquid system can be expanded as a polynomial based on number density, then Equation (18) takes the following form:

$$g_1(r_{ij}, T) = \int_V e^{-\varepsilon_{ii}/kT}dr \times \int_V e^{-\varepsilon_{jj}/kT}dr - \int_V e^{-\varepsilon_{ii}/kT}dr - \int_V e^{-\varepsilon_{jj}/kT}dr + \left(\int_V dr_{ii} + \int_V dr_{jj}\right) \tag{20}$$

Suppose that the subradial distribution function $g_1(r_{ij}, T)$ of the principal RDF $g_{ij}(r, \rho_0, T)$ conforms to $\rho_0 \to 0$, i.e., $i - j$ is the primary RDF, then $i - i$ and $j - j$ conform to $\rho_0 \to 0$ and the subradial distribution function is:

$$g_{ii}(r) = \exp\left(-\frac{\varepsilon_{ii}}{kT}\right) \quad g_{jj}(r) = \exp\left(-\frac{\varepsilon_{jj}}{kT}\right) \tag{21}$$

Substituting Equation (21) into Equation (20), we obtain:

$$g_1(r_{ij}, T) = \int_V g_{ii}(r)dr \times \int_V g_{jj}(r)dr - \int_V g_{ii}(r)dr - \int_V g_{jj}(r)dr + \left( \int_V dr_{ii} + \int_V dr_{jj} \right) \quad (22)$$

Then substituting Equation (22) into Equation (18), we obtain:

$$
\begin{aligned}
g_{ij}(r, \rho_0, T) &= e^{-\varepsilon_{ij}/kT} \exp\left\{ \rho_0 \left[ \begin{array}{c} \int_V g_{ii}(r)dr \times \int_V g_{jj}(r)dr - \int_V g_{ii}(r)dr \\ -\int_V g_{jj}(r)dr + (\int_V dr_{ii} + \int_V dr_{jj}) \end{array} \right] \right\} \\
&= e^{-\varepsilon_{ij}/kT} \left\{ 1 + \rho_0 \left[ \begin{array}{c} \int_V g_{ii}(r)dr \times \int_V g_{jj}(r)dr - \int_V g_{ii}(r)dr \\ -\int_V g_{jj}(r)dr + (\int_V dr_{ii} + \int_V dr_{jj}) \end{array} \right] \right\}
\end{aligned}
\quad (23)
$$

The relation between the RDF and potential energy can be obtained using Equation (23):

$$-\frac{\varepsilon_{ij}}{kT} = \ln \frac{g_{ij}(r, \rho_0, T)}{\left\{ 1 + \rho_0 \left[ \begin{array}{c} \int_V g_{ii}(r)dr \times \int_V g_{jj}(r)dr - \int_V g_{ii}(r)dr \\ -\int_V g_{jj}(r)dr + (\int_V dr_{ii} + \int_V dr_{jj}) \end{array} \right] \right\}} \quad (24)$$

Pair potential energy between molecules can be accurately calculated using the RDF. This function represents the ratio of the probability of finding another atom at a distance $r$ to the random distribution [16]. In the double distribution function, for a system with $N$ atoms and volume $V$, the probability of a atom appearing in the element $dr_i$ is $(N/V)dr_i$, the probability of an atom appearing at the distance $dr_j$ is $(N/V)dr_j$, and the probability of atomic pairs appearing at a distance $r$ is $(N/V)^2 dr_i dr_j$. The double distribution function is given as follows:

$$p^{(2)}(r)dr_i dr_j = \left(\frac{N}{V}\right)^2 dr_i dr_j \quad (25)$$

In the system, the average potential energy $\varepsilon$ between each atom is:

$$\varepsilon = \iint_V \varepsilon_{ij}(r) p^{(2)}(r) dr_i dr_j \quad (26)$$

However, in the RDF of binary systems, the probability of having atom $i$ in $dr_i$ at $r_i$ and atom $j$ in $dr_j$ at $r_j$ is $p^{(2)}(r)dr_i dr_j$. The potential energy is $\varepsilon$, and the average value of $\varepsilon$ is the sum of all possible times of the probabilities:

$$
\begin{aligned}
\overline{\varepsilon_{ij}} &= \frac{1}{V^2} \iint_V \varepsilon_{ij}(r) g_{ij}(r) dr_i dr_j = \frac{1}{V^2} \int_V dr_i \int \varepsilon_{ij}(r) g_{ij}(r) dr_j = \frac{1}{V} \int \varepsilon_{ij}(r) g_{ij}(r) 4\pi r^2 dr \\
&= \frac{4\pi \int \varepsilon_{ij}(r) g_{ij}(r) r^2 dr}{4\pi \int r^2 dr} = \frac{\int \varepsilon_{ij}(r) g_{ij}(r) r^2 dr}{\int r^2 dr}
\end{aligned}
\quad (27)
$$

Substituting Equation (24) into Equation (27) yields the potential energy between atoms:

$$-\frac{\overline{\varepsilon_{ij}}}{kT} = \frac{\int g_{ij}(r)r^2 \left\{ \ln \frac{g_{ij}(r)}{1 + \rho_0 \left[ \begin{array}{c} \int_V g_{ii}(r)dr \times \int_V g_{jj}(r)dr - \int_V g_{ii}(r)dr \\ -\int_V g_{jj}(r)dr + (\int_V dr_{ii} + \int_V dr_{jj}) \end{array} \right]} \right\} dr}{\int r^2 dr} \quad (28)$$

The peak value of the RDF signifies the disparity between the local and bulk molar fractions. As the mole fraction increases, the contributions of the second and third RDF peaks diminish while the contribution of the first peak amplifies. The pair potential energy is then calculated by using Equation (28) and the area of the first peak of the skewed RDF.

The area under the first peak was computed through graphical integration, as depicted in Figure 4. Notably, this approach differs from Wang's utilization of the L-PPDF mathematical form with Gaussian fitting, which relies on fitting parameters u and v [17,18]. It is important to emphasize that this study does not incorporate any fitting parameters. The RDF used in this study is defined by three key coordinates: $r_0$ (which represents the starting point of $g(r)$ before reaching zero), $r_m$ (the transverse coordinate of the first peak of $g(r)$), and $r_1$ (the transverse coordinate of the first valley of $g(r)$). The asymmetric method of calculating the RDF involves integrating the area between $r_0$ and $r_1$, while the symmetric method involves integrating twice the area between $r_0$ and $r_m$ (Figure 4). The trapezoidal method [19] is used to compute these areas (Equation (29)). Table 2 lists the references for the partial RDFs of 36 binary liquid alloys.

$$\int_{r1}^{r0} g(r)dr \approx \frac{b-a}{2N}\sum_{n=1}^{N}[g(r_n)+g(r_{n+1})] = \frac{b-a}{2N}[g(r)+2g(r_2)+\ldots\ldots+2g(r_N)+g(r_{N+1})] \tag{29}$$

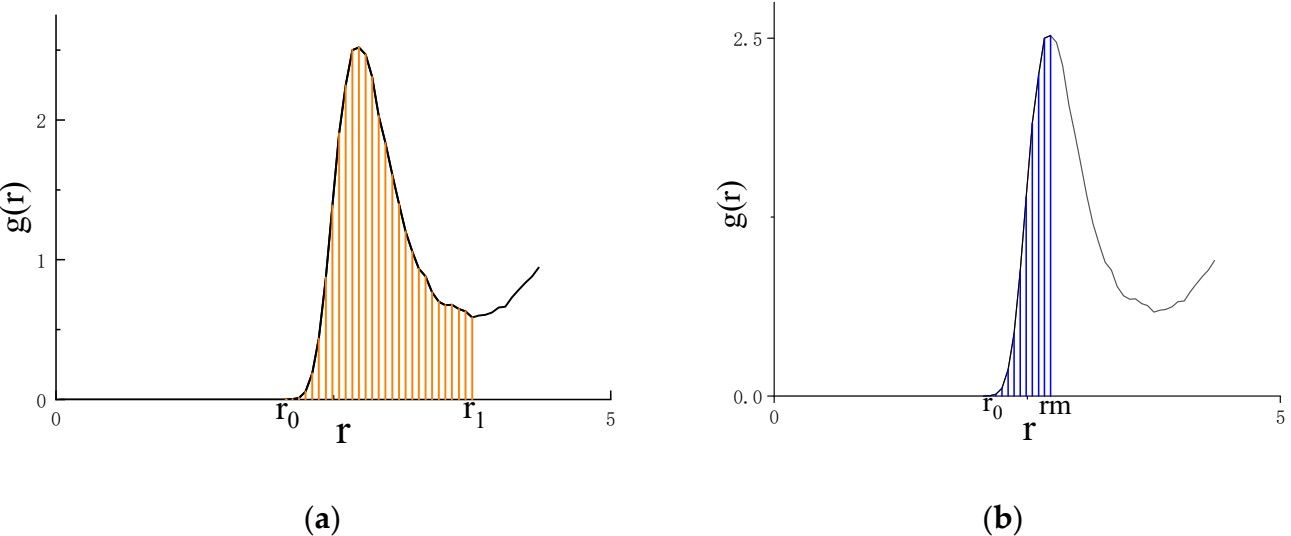

**Figure 4.** (**a**) Schematic diagram of the asymmetric method of graphical integration of a radial distribution function, (**b**) Schematic diagram of the symmetric method of graphical integration of a radial distribution function.

**Table 2.** Partial radial distribution functions for 36 binary liquid alloys in literatures.

| System | Co-Ni [20] | Al-Zn [21] | Cu-Ni [22] | Al-Ni [23] | Cu-Fe [24] | Ge-Sn [25] | Ag-Cu [23] | Pb-Sb [26] | Al-Si [27] | Al-Co [28] |
|---|---|---|---|---|---|---|---|---|---|---|
| System | Li-Mg [29] | Sb-Sn [30] | Cu-Zr [31] | K-Na [32] | Pb-Sn [33] | Al-Mg [34] | Cs-K [35] | Li-Na [36] | Au-Cu [37] | Al-Li [38] |
| System | Nb-Zr [39] | Ni-Pd [40] | Cu-Mg [41] | Al-Ca [42] | Ni-Zr [43] | Al-Sn [44] | Al-Cu [45] | Nb-Ni [46] | Cu-Sn [47] | Au-Si [48] |
| System | Li-Sn [49] | Fe-Si [50] | Ag-In [51] | Ge-Te [52] | Al-Au [53] | Cu-Sb [54] | | | | |

The RSM has a tunable parameter $\Omega_{ij}$. The average coordination number $\overline{Z}$ is obtained using the pure coordination number of the two components. Additionally, the temperature $T$ documented in the literature and the calculated parameter $\Omega_{ij}$ can be referred to calculate the parameter $\Omega_{ij}'$ at the desired temperature $T'$.

$$\overline{Z} = \frac{1}{2}(Z_i + Z_j) \tag{30}$$

$$\Omega_{ij} = kT\left\{\overline{Z}\left[\overline{\varepsilon_{ij}} - \frac{1}{2}(\overline{\varepsilon_{ii}} + \overline{\varepsilon_{jj}})\right]\right\} \tag{31}$$

$$T\ln\Omega_{ij} = T'\ln\Omega_{ij}' \tag{32}$$

The parameters $A_{ij}$ and $A_{ji}$ of Wilson equation are given. Additionally, the values of parameters $A_{ij}'$ and $A_{ij}'$ at the desired temperature $T'$ can be obtained using the temperature $T$ from the literature that was employed to calculate the corresponding parameters $A_{ij}$ and $A_{ji}$.

$$A_{ij} = \frac{V_i}{V_j}\exp\left(-\frac{\overline{\varepsilon_{ij}} - \overline{\varepsilon_{jj}}}{RT}\right) \quad A_{ji} = \frac{V_j}{V_i}\exp\left(-\frac{\overline{\varepsilon_{ji}} - \overline{\varepsilon_{ii}}}{RT}\right) \tag{33}$$

$$T\ln A_{ij} = T'\ln A_{ij}' \quad T\ln A_{ji} = T'\ln A_{ij}' \tag{34}$$

The NRTL has two parameters $\tau_{ij}$ and $\tau_{ji}$. These parameters obtained using the temperature $T$ mentioned in the literature are employed to calculate the parameters $\tau_{ij}'$ and $\tau_{ij}'$, respectively, at the required temperature $T'$.

$$\tau_{ij} = -\frac{\overline{\varepsilon_{ij}} - \overline{\varepsilon_{jj}}}{kT} \quad \tau_{ji} = -\frac{\overline{\varepsilon_{ji}} - \overline{\varepsilon_{ii}}}{kT} \tag{35}$$

$$T\ln\tau_{ij} = T'\ln\tau_{ij}' \quad T\ln\tau_{ji} = T'\ln\tau_{ij}' \tag{36}$$

The MIVM involves parameters $B_{ij}$ and $B_{ji}$, representing the interaction parameters for $i$–$j$ and $j$–$i$ interactions, respectively. Using $B_{ij}$ and $B_{ji}$ values obtained at temperature $T$, the corresponding parameters $B_{ij}'$ and $B_{ij}'$ can be calculated at the desired temperature $T'$.

$$B_{ij} = \exp\left(-\frac{\overline{\varepsilon_{ij}} - \overline{\varepsilon_{jj}}}{kT}\right) \quad B_{ji} = \exp\left(-\frac{\overline{\varepsilon_{ji}} - \overline{\varepsilon_{ii}}}{kT}\right) \tag{37}$$

$$T\ln B_{ij} = T'\ln B_{ij}' \quad T\ln B_{ji} = T'\ln B_{ij}' \tag{38}$$

## 4. Result Analysis

### 4.1. Asymmetric Method for Calculating the RDF

The asymmetric method uses the area between $r_0$ and $r_1$ presented in Figure 4 and Equation (28) to determine the parameters for each model, see Table 3. Table 4 demonstrates that among the first 12 systems, the RSM performs better than other models for six binary liquid alloys. The average standard deviation (SD) is 0.027, and the average relative deviation (ARD) is 7.7%. In the case of the 12 binary liquid alloys with moderate symmetry, the RSM outperforms the other three models in six systems, resulting in an average SD of 0.059 and an average ARD of 13.83%. MIVM also performs well, with an average SD of 0.091 and an average ARD of 25.7%. For the 12 binary liquid alloys with low symmetry, the RSM outperforms the other three models in three systems, yielding an average SD of 0.101 and an average ARD of 39.5%. Additionally, the MIVM outperforms the other three models in five binary liquid alloys when the binary liquid alloys have even lower symmetry, with an average SD of 0.131 and an average ARD of 45.3%. Considering the average performance across all 36 binary liquid alloys, the Wilson and NRTL models exhibit the poorest performance, displaying larger SD and ARD values than the other models. As shown in Figure 5, the SD and ARD values generally increase with decreasing symmetry, albeit not significantly. Further analysis of high, medium, and low-symmetry binary liquid alloys reveals a strong correlation between the RSM and symmetry.

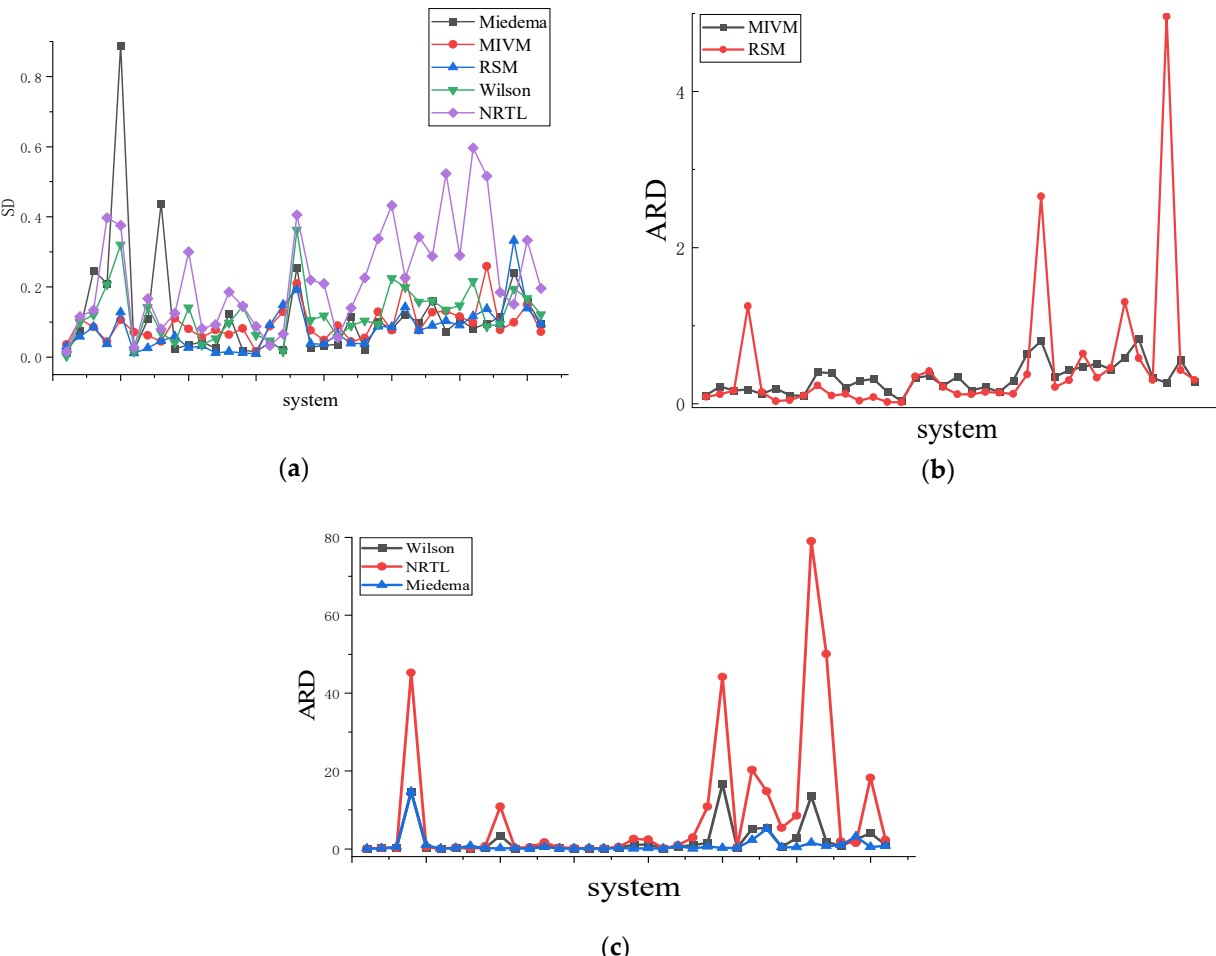

**Figure 5.** (**a**) is the SD of 36 binary liquid alloys, (**b**) is the ARD of MIVM and RSM models of 36 binary liquid alloys, (**c**) is the ARD of the Wilson equation, NRTL equation, and Miedema of 36 binary liquid alloys.

**Table 3.** Parameters of four models of the asymmetric method.

| System | MIVM | | RSM | Wilson | | NRTL | |
|---|---|---|---|---|---|---|---|
| | $B_{ij}'$ | $B_{ij}'$ | $\Omega_{ij}' = \Omega_{ij}'$ | $A_{ij}'$ | $A_{ij}'$ | $\tau_{ij}'$ | $\tau_{ij}'$ |
| Co-Ni | 0.964 | 0.977 | 0.343 | 0.964 | 0.977 | −0.036 | −0.023 |
| Al-Zn | 1.246 | 0.743 | 0.423 | 1.166 | 0.795 | 0.220 | −0.297 |
| Cu-Ni | 1.086 | 0.863 | 0.374 | 0.999 | 0.938 | 0.082 | −0.147 |
| Al-Ni | 1.493 | 1.649 | −5.203 | 0.991 | 2.485 | 0.401 | 0.500 |
| Cu-Fe | 0.943 | 0.805 | 1.512 | 0.943 | 0.805 | −0.059 | −0.217 |
| Ge-Sn | 0.740 | 1.273 | 0.266 | 1.214 | 0.776 | −0.302 | 0.241 |
| Ag-Cu | 0.675 | 1.194 | 1.225 | 0.471 | 1.709 | −0.394 | 0.177 |
| Pb-Sb | 1.558 | 0.517 | 1.046 | 1.457 | 0.553 | 0.443 | −0.660 |
| Al-Si | 1.405 | 1.077 | −1.856 | 1.226 | 1.235 | 0.340 | 0.074 |
| Al-Co | 1.112 | 1.884 | −4.213 | 0.802 | 2.612 | 0.106 | 0.633 |
| Li-Mg | 1.357 | 0.914 | −1.095 | 1.469 | 0.844 | 0.305 | −0.090 |
| Sb-Sn | 0.769 | 1.545 | −0.847 | 0.735 | 1.618 | −0.262 | 0.435 |
| Cu-Zr | 0.908 | 1.669 | −2.277 | 1.782 | 0.851 | −0.096 | 0.512 |
| K-Na | 0.700 | 1.176 | 1.018 | 0.366 | 2.252 | −0.357 | 0.162 |
| Pb-Sn | 1.044 | 0.912 | 0.267 | 0.932 | 1.021 | 0.043 | −0.092 |
| Al-Mg | 0.820 | 1.123 | 0.459 | 1.162 | 0.793 | −0.198 | 0.116 |

**Table 3.** *Cont.*

| System | MIVM | | RSM | Wilson | | NRTL | |
|---|---|---|---|---|---|---|---|
| | $B_{ij}'$ | $B_{ij}'$ | $\Omega_{ij}' = \Omega_{ij}'$ | $A_{ij}'$ | $A_{ij}'$ | $\tau_{ij}'$ | $\tau_{ij}'$ |
| Cs-K | 1.204 | 0.635 | 1.310 | 0.801 | 0.955 | 0.185 | −0.454 |
| Li-Na | 0.764 | 0.999 | 1.347 | 1.416 | 0.539 | −0.270 | −0.001 |
| Au-Cu | 1.163 | 1.444 | −2.877 | 0.812 | 2.068 | 0.151 | 0.368 |
| Al-Li | 1.136 | 1.379 | −2.356 | 1.485 | 1.055 | 0.128 | 0.321 |
| Nb-Zr | 0.814 | 1.255 | −0.110 | 1.048 | 0.975 | −0.206 | 0.227 |
| Ni-Pd | 1.168 | 1.087 | −1.307 | 1.590 | 0.798 | 0.153 | 0.080 |
| Cu-Mg | 1.312 | 1.258 | −2.781 | 2.575 | 0.641 | 0.272 | 0.230 |
| Al-Ca | 2.202 | 1.259 | −5.761 | 5.826 | 0.476 | 0.789 | 0.230 |
| Ni-Zr | 1.522 | 1.730 | −5.373 | 3.247 | 0.811 | 0.420 | 0.548 |
| Al-Sn | 0.624 | 1.440 | 0.598 | 1.024 | 0.877 | −0.471 | 0.365 |
| Al-Cu | 1.568 | 1.530 | −5.209 | 2.192 | 1.152 | 0.476 | 0.450 |
| Nb-Ni | 1.772 | 1.071 | −3.558 | 1.070 | 1.774 | 0.572 | 0.069 |
| Cu-Sn | 1.987 | 0.849 | −2.904 | 0.874 | 1.932 | 0.687 | −0.163 |
| Au-Si | 1.224 | 1.684 | −3.129 | 1.034 | 1.993 | 0.202 | 0.521 |
| Li-Sn | 3.396 | 2.186 | −10.225 | 4.263 | 1.742 | 1.223 | 0.782 |
| Fe-Si | 5.678 | 1.776 | −9.822 | 4.695 | 2.148 | 1.737 | 0.574 |
| Ag-In | 1.595 | 0.925 | −2.227 | 2.476 | 0.596 | 0.467 | −0.078 |
| Ge-Te | 0.409 | 1.891 | 1.285 | 0.912 | 0.848 | −0.894 | 0.637 |
| Al-Au | 1.680 | 1.660 | −5.740 | 2.287 | 1.219 | 0.519 | 0.507 |
| Cu-Sb | 1.804 | 0.881 | −2.315 | 0.757 | 2.098 | 0.590 | −0.127 |

**Table 4.** Deviations and relative errors of five models in the asymmetric method.

| System | Miedema | | MIVM | | RSM | | Wilson | | NRTL | |
|---|---|---|---|---|---|---|---|---|---|---|
| | SD | ARD/% | SD | ARD/% | SD | ARD/% | SD | ARD/% | SD | ARD/% |
| Co-Ni | 0.013 | 3.852 | 0.036 | 10.8 | 0.029 | 8.8 | 0.003 | 0.8 | 0.015 | 4.4 |
| Al-Zn | 0.075 | 15.536 | 0.108 | 22 | 0.058 | 12.1 | 0.100 | 20.5 | 0.116 | 23.7 |
| Cu-Ni | 0.247 | 49.697 | 0.085 | 16.5 | 0.085 | 16.5 | 0.120 | 23.1 | 0.134 | 25.7 |
| Al-Ni | 0.210 | 1467.533 | 0.044 | 18.3 | 0.037 | 125 | 0.208 | 1472 | 0.398 | 4520 |
| Cu-Fe | 0.889 | 93.800 | 0.106 | 12.5 | 0.128 | 15.1 | 0.321 | 37.5 | 0.375 | 44 |
| Ge-Sn | 0.019 | 5.384 | 0.071 | 19.4 | 0.011 | 3.1 | 0.016 | 4.1 | 0.027 | 7.5 |
| Ag-Cu | 0.108 | 19.924 | 0.062 | 10.8 | 0.025 | 4.6 | 0.143 | 26.1 | 0.167 | 30.5 |
| Pb-Sb | 0.438 | 79.951 | 0.044 | 9.9 | 0.047 | 10.5 | 0.070 | 15.9 | 0.080 | 18.1 |
| Al-Si | 0.023 | 10.017 | 0.110 | 40.5 | 0.058 | 23.2 | 0.041 | 18.5 | 0.125 | 63.3 |
| Al-Co | 0.034 | 10.163 | 0.080 | 39.2 | 0.027 | 10.5 | 0.142 | 341 | 0.300 | 1085 |
| Li-Mg | 0.043 | 17.882 | 0.056 | 20.7 | 0.032 | 12.3 | 0.035 | 14.2 | 0.082 | 34.9 |
| Sb-Sn | 0.025 | 6.367 | 0.078 | 29.3 | 0.012 | 3.6 | 0.054 | 24.5 | 0.093 | 44.2 |
| Cu-Zr | 0.124 | 53.790 | 0.064 | 31.6 | 0.015 | 8.4 | 0.098 | 74.4 | 0.185 | 162 |
| K-Na | 0.019 | 3.619 | 0.081 | 15.3 | 0.012 | 2 | 0.144 | 27.8 | 0.145 | 28 |
| Pb-Sn | 0.015 | 1.847 | 0.016 | 3.2 | 0.009 | 1.7 | 0.063 | 14.3 | 0.087 | 19.6 |
| Al-Mg | 0.042 | 14.076 | 0.087 | 32.9 | 0.092 | 34.9 | 0.048 | 17 | 0.033 | 11.2 |
| Cs-K | 0.020 | 4.475 | 0.129 | 36 | 0.148 | 41.7 | 0.016 | 3.3 | 0.066 | 16.9 |
| Li-Na | 0.255 | 24.451 | 0.210 | 22.9 | 0.192 | 21.3 | 0.364 | 40.5 | 0.405 | 45.2 |
| Au-Cu | 0.028 | 7.697 | 0.076 | 34.5 | 0.038 | 12.1 | 0.105 | 101 | 0.219 | 255 |
| Al-Li | 0.034 | 17.993 | 0.048 | 17 | 0.037 | 12.2 | 0.118 | 115 | 0.209 | 236 |
| Nb-Zr | 0.036 | 8.347 | 0.089 | 21.1 | 0.066 | 15.1 | 0.058 | 13 | 0.056 | 12.4 |
| Ni-Pd | 0.115 | 72.225 | 0.044 | 15 | 0.039 | 14.1 | 0.090 | 53.7 | 0.140 | 91 |
| Cu-Mg | 0.022 | 8.778 | 0.054 | 30.2 | 0.038 | 12.4 | 0.104 | 103 | 0.226 | 286 |
| Al-Ca | 0.100 | 56.098 | 0.129 | 64.3 | 0.087 | 37.4 | 0.094 | 144 | 0.338 | 1081 |
| Ni-Zr | 0.080 | 20.755 | 0.076 | 80.2 | 0.085 | 266 | 0.225 | 1661 | 0.433 | 4415 |
| Al-Sn | 0.122 | 18.737 | 0.226 | 34.4 | 0.142 | 21.3 | 0.199 | 30.4 | 0.226 | 34.5 |
| Al-Cu | 0.099 | 228.621 | 0.076 | 43 | 0.075 | 30.1 | 0.158 | 516 | 0.343 | 2031 |
| Nb-Ni | 0.159 | 517.211 | 0.128 | 47.2 | 0.089 | 64 | 0.161 | 540 | 0.288 | 1478 |
| Cu-Sn | 0.073 | 38.233 | 0.131 | 51.2 | 0.103 | 33.3 | 0.135 | 48 | 0.524 | 544 |

**Table 4.** *Cont.*

| System | Miedema | | MIVM | | RSM | | Wilson | | NRTL | |
|---|---|---|---|---|---|---|---|---|---|---|
| | SD | ARD/% | SD | ARD/% | SD | ARD/% | SD | ARD/% | SD | ARD/% |
| Au-Si | 0.106 | 43.309 | 0.116 | 44 | 0.090 | 45.6 | 0.147 | 279 | 0.290 | 847 |
| Li-Sn | 0.081 | 159.055 | 0.098 | 59.1 | 0.117 | 130 | 0.216 | 1347 | 0.596 | 7906 |
| Fe-Si | 0.095 | 67.563 | 0.259 | 82.9 | 0.137 | 58.3 | 0.088 | 186 | 0.516 | 5003 |
| Ag-In | 0.114 | 97.817 | 0.077 | 32.8 | 0.095 | 30.3 | 0.095 | 76 | 0.185 | 191 |
| Ge-Te | 0.240 | 319.161 | 0.099 | 27.3 | 0.333 | 496 | 0.195 | 239 | 0.150 | 148 |
| Al-Au | 0.159 | 53.722 | 0.149 | 55.9 | 0.139 | 42.7 | 0.168 | 405 | 0.333 | 1822 |
| Cu-Sb | 0.095 | 80.468 | 0.072 | 28 | 0.096 | 30.4 | 0.122 | 104 | 0.196 | 229 |
| Ave | 0.121 | 102.7 | 0.095 | 32.2 | 0.078 | 47.4 | 0.124 | 226 | 0.225 | 911 |

$\text{SD} = \sqrt{\frac{\sum(a_{pre} - a_{\exp})^2}{N}}$; $\text{ARD} = \frac{1}{N}\sum\left|\frac{a_{pre} - a_{\exp}}{a_{\exp}}\right| \times 100\%$. $a_{pre}$-calculated value of the activity; $a_{\exp}$ [55–59]-experimental activity value.

### 4.2. Symmetric Method for Calculating the RDF

The methodology based on symmetry involves deriving parameters for each model using the region from $r_0$ to $r_1$ (Figure 4) and Equation (28), see Table 5. Analysis of data presented in Table 6 reveals that among the 12 systems characterized by high symmetry, the MIVM outperforms the other three models with an average SD of 0.127 and an average ARD of 30.5%. The RSM exhibits an average SD of 0.111 and an average ARD of 44.6%. In the 12 systems with medium symmetry, the Miedema outperforms the other three models, yielding a average SD of 0.067 and an ARD of 22.8%. By contrast, the MIVM exhibits a average SD of 0.118 and an ARD of 36.7%, and the RSM exhibits a average SD of 0.088 and an ARD of 32.8%. For the 12 systems with low symmetry, the Miedema surpasses the other three models. The RSM exhibits an average SD of 0.116 and an average ARD of 70.2%. Each model exhibits distinct performance characteristics depending on the system representation. As shown in Figure 6, the data comparison indicates an increasing trend in ARD values with decreasing symmetry.

**Table 5.** Parameters of four models of the symmetric method.

| System | MIVM | | RSM | Wilson | | NRTL | |
|---|---|---|---|---|---|---|---|
| | $B_{ij}'$ | $B_{ji}'$ | $\Omega_{ij}'=\Omega_{ji}'$ | $A_{ij}'$ | $A_{ji}'$ | $\tau_{ij}'$ | $\tau_{ji}'$ |
| Co-Ni | 1.089 | 0.900 | 0.118 | 1.089 | 0.900 | 0.085 | −0.106 |
| Al-Zn | 1.317 | 0.938 | −1.162 | 1.232 | 1.003 | 0.275 | −0.064 |
| Cu-Ni | 1.391 | 0.877 | −1.138 | 1.280 | 0.953 | 0.330 | −0.132 |
| Al-Ni | 1.155 | 2.099 | −5.114 | 0.766 | 3.163 | 0.144 | 0.742 |
| Cu-Fe | 0.890 | 1.145 | −0.103 | 0.890 | 1.145 | −0.117 | 0.135 |
| Ge-Sn | 0.562 | 1.573 | 0.545 | 0.923 | 0.958 | −0.576 | 0.453 |
| Ag-Cu | 1.309 | 0.582 | 1.538 | 0.914 | 0.833 | 0.269 | −0.542 |
| Pb-Sb | 1.231 | 0.810 | 0.017 | 1.151 | 0.866 | 0.207 | −0.211 |
| Al-Si | 1.534 | 1.140 | −2.499 | 1.338 | 1.307 | 0.428 | 0.131 |
| Al-Co | 0.830 | 1.774 | −2.218 | 0.551 | 2.673 | −0.186 | 0.573 |
| Li-Mg | 1.087 | 0.988 | −0.363 | 1.177 | 0.912 | 0.083 | −0.012 |
| Sb-Sn | 0.825 | 1.576 | −1.288 | 0.788 | 1.650 | −0.192 | 0.455 |
| Cu-Zr | 1.206 | 1.206 | 1.206 | 1.206 | 1.206 | 1.206 | 1.206 |
| K-Na | 0.562 | 1.364 | 1.390 | 0.293 | 2.613 | −0.577 | 0.311 |
| Pb-Sn | 1.601 | 0.830 | −1.549 | 1.430 | 0.929 | 0.471 | −0.186 |
| Al-Mg | 0.808 | 1.126 | 0.528 | 1.144 | 0.796 | −0.213 | 0.119 |
| Cs-K | 1.642 | 0.660 | −0.391 | 1.092 | 0.992 | 0.496 | −0.416 |
| Li-Na | 0.728 | 1.069 | 1.245 | 1.350 | 0.577 | −0.317 | 0.067 |
| Au-Cu | 2.423 | 0.515 | −1.229 | 1.692 | 0.737 | 0.885 | −0.664 |
| Al-Li | 0.813 | 1.374 | −0.580 | 1.062 | 1.051 | −0.208 | 0.318 |

**Table 5.** *Cont.*

| System | MIVM | | RSM | Wilson | | NRTL | |
|---|---|---|---|---|---|---|---|
| | $B_{ij}'$ | $B_{ji}'$ | $\Omega_{ij}'=\Omega_{ji}'$ | $A_{ij}'$ | $A_{ji}'$ | $\tau_{ij}'$ | $\tau_{ji}'$ |
| Nb-Zr | 0.671 | 1.414 | 0.279 | 0.864 | 1.098 | −0.399 | 0.346 |
| Ni-Pd | 1.174 | 1.105 | 11.250 | −1.445 | 1.599 | 0.811 | 0.159 |
| Cu-Mg | 0.882 | 1.538 | −1.689 | 1.731 | 0.783 | −0.126 | 0.430 |
| Al-Ca | 2.423 | 0.515 | −1.229 | 1.692 | 0.737 | 0.885 | −0.664 |
| Ni-Zr | 2.150 | 4.152 | −12.151 | 4.587 | 1.947 | 0.766 | 1.424 |
| Al-Sn | 0.649 | 1.393 | 0.566 | 1.065 | 0.849 | −0.433 | 0.332 |
| Al-Cu | 1.963 | 1.534 | −6.647 | 1.472 | 2.180 | 0.714 | 0.453 |
| Nb-Ni | 3.602 | 1.064 | −7.456 | 2.174 | 1.763 | 1.281 | 0.062 |
| Cu-Sn | 4.758 | 0.617 | −5.972 | 2.091 | 1.403 | 1.560 | −0.484 |
| Au-Si | 1.438 | 0.875 | −0.992 | 1.214 | 1.036 | 0.363 | −0.134 |
| Li-Sn | 2.077 | 2.672 | −8.741 | 2.608 | 2.129 | 0.731 | 0.983 |
| Fe-Si | 1.534 | 1.843 | −4.414 | 1.268 | 2.228 | 0.428 | 0.611 |
| Ag-In | 1.066 | 1.467 | −2.560 | 1.654 | 0.946 | 0.064 | 0.384 |
| Ge-Te | 0.636 | 3.025 | −3.272 | 1.419 | 1.356 | −0.452 | 1.107 |
| Al-Au | 1.903 | 0.844 | −2.370 | 0.799 | 2.011 | 0.643 | −0.169 |
| Cu-Sb | 1.743 | 1.017 | −3.207 | 1.743 | 1.017 | 0.556 | 0.017 |

**Table 6.** Deviations and relative errors of five models in the symmetric method.

| System | Miedema | | MIVM | | RSM | | Wilson | | NRTL | |
|---|---|---|---|---|---|---|---|---|---|---|
| | SD | ARD/% | SD | ARD/% | SD | ARD/% | SD | ARD/% | SD | ARD/% |
| Co-Ni | 0.013 | 3.852 | 0.002 | 0.4 | 0.004 | 1.1 | 0.007 | 2.1 | 0.011 | 3.3 |
| Al-Zn | 0.075 | 15.536 | 0.233 | 45.8 | 0.201 | 40 | 0.128 | 26 | 0.085 | 17.5 |
| Cu-Ni | 0.247 | 49.697 | 0.252 | 46.9 | 0.220 | 41.3 | 0.147 | 28.3 | 0.107 | 20.8 |
| Al-Ni | 0.210 | 1467.533 | 0.081 | 31.7 | 0.039 | 133 | 0.257 | 2130 | 0.335 | 3370 |
| Cu-Fe | 0.889 | 93.800 | 0.365 | 42.8 | 0.359 | 42 | 0.351 | 41.2 | 0.347 | 40.7 |
| Ge-Sn | 0.019 | 5.384 | 0.135 | 35.4 | 0.045 | 13.2 | 0.007 | 1.6 | 0.039 | 10.8 |
| Ag-Cu | 0.108 | 19.924 | 0.063 | 10.2 | 0.087 | 16.3 | 0.115 | 21.2 | 0.175 | 31.8 |
| Pb-Sb | 0.438 | 79.951 | 0.054 | 17.9 | 0.164 | 62.6 | 0.038 | 13.3 | 0.004 | 0.9 |
| Al-Si | 0.023 | 10.017 | 0.147 | 50.8 | 0.089 | 34.2 | 0.027 | 12.1 | 0.142 | 72.6 |
| Al-Co | 0.034 | 10.163 | 0.074 | 39.6 | 0.070 | 130 | 0.161 | 411 | 0.258 | 852 |
| Li-Mg | 0.043 | 17.882 | 0.020 | 7.9 | 0.026 | 10.2 | 0.052 | 21.7 | 0.067 | 28.5 |
| Sb-Sn | 0.025 | 6.367 | 0.100 | 36.5 | 0.029 | 11.3 | 0.046 | 20.7 | 0.103 | 49.6 |
| Cu-Zr | 0.124 | 53.790 | 0.065 | 33 | 0.026 | 14.4 | 0.084 | 62.1 | 0.193 | 171 |
| K-Na | 0.019 | 3.619 | 0.130 | 24.3 | 0.076 | 15.2 | 0.154 | 29.6 | 0.156 | 29.9 |
| Pb-Sn | 0.015 | 1.847 | 0.244 | 51.4 | 0.194 | 42 | 0.105 | 23.7 | 0.049 | 11 |
| Al-Mg | 0.042 | 14.076 | 0.095 | 36.2 | 0.101 | 38.4 | 0.049 | 17.5 | 0.031 | 10.7 |
| Cs-K | 0.020 | 4.475 | 0.132 | 32.3 | 0.074 | 18.8 | 0.046 | 11.5 | 0.036 | 8.9 |
| Li-Na | 0.255 | 24.451 | 0.244 | 26.7 | 0.211 | 23.4 | 0.363 | 40.4 | 0.404 | 45.1 |
| Au-Cu | 0.028 | 7.697 | 0.117 | 40.7 | 0.067 | 58.8 | 0.135 | 137 | 0.168 | 181 |
| Al-Li | 0.034 | 17.993 | 0.093 | 86.2 | 0.110 | 106 | 0.151 | 156 | 0.171 | 182 |
| Nb-Zr | 0.036 | 8.347 | 0.092 | 21.6 | 0.032 | 6.3 | 0.052 | 11.5 | 0.065 | 14.7 |
| Ni-Pd | 0.115 | 72.225 | 0.046 | 15.2 | 0.036 | 11.3 | 0.088 | 52.4 | 0.143 | 93.2 |
| Cu-Mg | 0.022 | 8.778 | 0.065 | 22.3 | 0.051 | 40 | 0.136 | 144 | 0.202 | 245 |
| Al-Ca | 0.100 | 56.098 | 0.089 | 50.3 | 0.075 | 20 | 0.107 | 175 | 0.301 | 901 |
| Ni-Zr | 0.080 | 20.755 | 0.248 | 75.6 | 0.098 | 35.8 | 0.140 | 86.2 | 0.573 | 6633.0 |
| Al-Sn | 0.122 | 18.737 | 0.216 | 32.8 | 0.147 | 21.9 | 0.200 | 30.6 | 0.224 | 34.3 |
| Al-Cu | 0.099 | 228.621 | 0.113 | 57.6 | 0.078 | 39.9 | 0.114 | 289 | 0.417 | 2930 |
| Nb-Ni | 0.159 | 517.211 | 0.216 | 67.7 | 0.104 | 49.9 | 0.121 | 302 | 0.363 | 2270 |
| Cu-Sn | 0.073 | 38.233 | 0.255 | 75.4 | 0.163 | 56.7 | 0.076 | 28 | 0.200 | 167 |
| Au-Si | 0.106 | 43.309 | 0.070 | 79 | 0.136 | 240 | 0.192 | 425 | 0.234 | 591 |
| Li-Sn | 0.081 | 159.055 | 0.150 | 86.9 | 0.116 | 187 | 0.244 | 1730 | 0.564 | 7420 |
| Fe-Si | 0.095 | 67.563 | 0.138 | 45.8 | 0.105 | 79.9 | 0.165 | 693 | 0.358 | 2900 |

**Table 6.** *Cont.*

| System | Miedema | | MIVM | | RSM | | Wilson | | NRTL | |
|---|---|---|---|---|---|---|---|---|---|---|
| | SD | ARD/% | SD | ARD/% | SD | ARD/% | SD | ARD/% | SD | ARD/% |
| Ag-In | 0.114 | 97.817 | 0.122 | 37.9 | 0.097 | 30.1 | 0.109 | 90.8 | 0.193 | 204 |
| Ge-Te | 0.240 | 319.161 | 0.245 | 73.9 | 0.114 | 29.1 | 0.135 | 118 | 0.211 | 268 |
| Al-Au | 0.159 | 53.722 | 0.149 | 55.9 | 0.139 | 42.7 | 0.168 | 405 | 0.333 | 1820 |
| Cu-Sb | 0.095 | 80.468 | 0.072 | 29.9 | 0.097 | 30 | 0.122 | 105 | 0.196 | 228 |
| Ave | 0.121 | 102.7 | 0.137 | 42.35 | 0.105 | 49.24 | 0.128 | 219.2 | 0.207 | 884.9 |

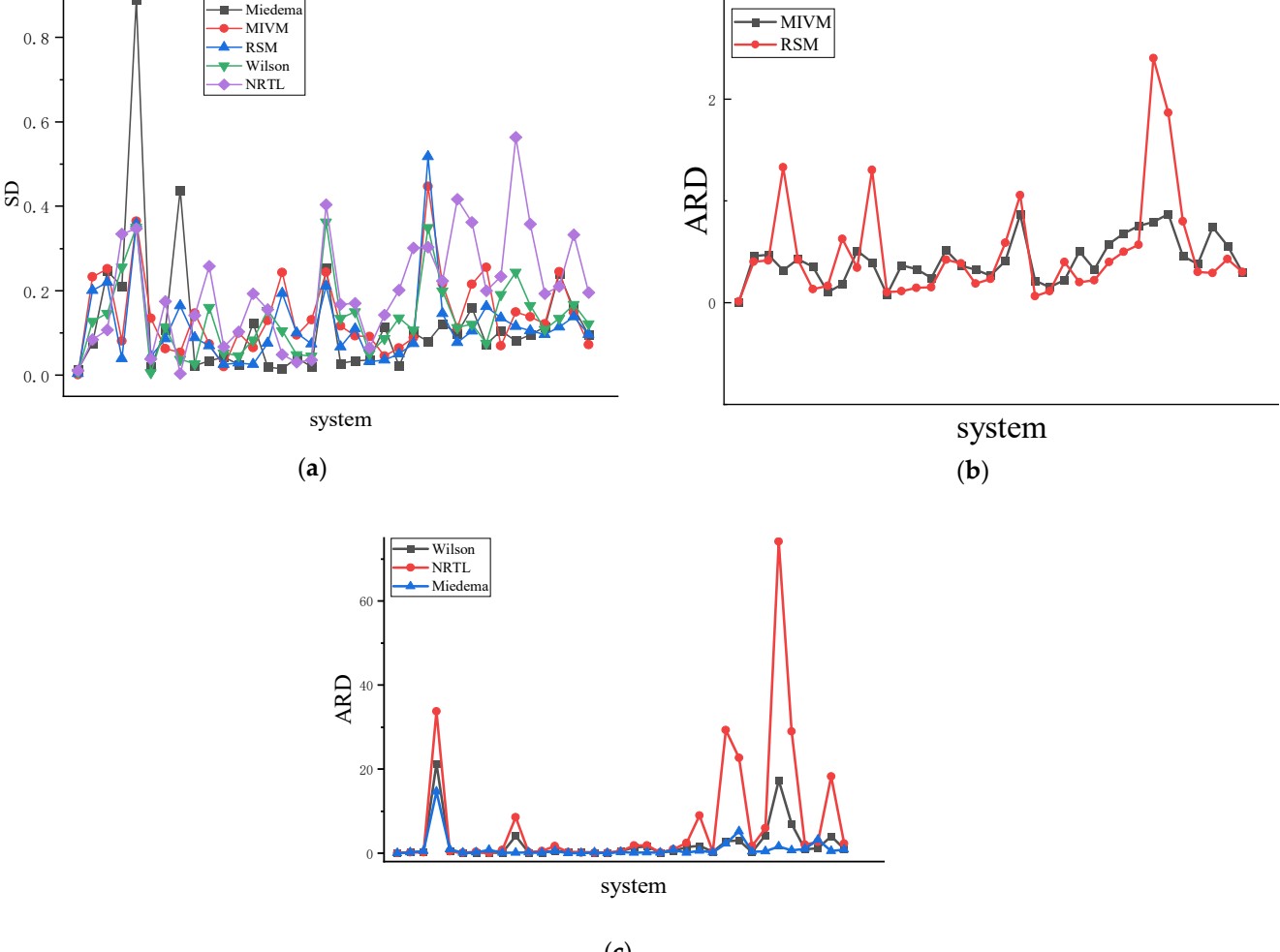

**Figure 6.** (**a**) is the SD of 36 binary liquid alloys, (**b**) is the ARD of MIVM and RSM models of 36 binary liquid alloys, (**c**) is the ARD of the Wilson equation, NRTL equation, and Miedema of 36 binary liquid alloys.

## 5. Conclusions

This study uses polynomials to describe the partial RDF in pure gases and extends this approach to binary liquid systems. The primary aim of this study is to characterize the atomic distribution and unravel interatomic interactions, which are essential for accurate thermodynamic calculations. The RDF exhibits irregularities when the symmetric method is used instead of the asymmetric one for RDF calculation. Notably, the estimation of binary liquid alloy activity favors using the asymmetric method, especially when considering the average results obtained from both methods for the 36 binary liquid alloy systems investigated in this study. We selected and compared five thermodynamic models based on

their symmetry degree. Data analysis reveals that the RSM exhibits the highest dependency on the symmetry degree. Conversely, the MIVM demonstrates superior adaptability to symmetric and asymmetric systems.

**Author Contributions:** D.T.: Theoretical guidance, review; J.H.: Conceptualization, Writing—original draft, Writing—review & editing. All authors have read and agreed to the published version of the manuscript.

**Funding:** This work was financially supported by the National Natural Science Foundation of China under Grant No. 51464022.

**Data Availability Statement:** Data available on request from the authors.

**Conflicts of Interest:** The authors declare that they have no known competing financial interest or personal relationship that could have appeared to influence the work reported in this paper.

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
