# Peer review of "Estimation of Two Component Activities of Binary Liquid Alloys by the Pair Potential Energy Containing a Polynomial of the Partial Radial Distribution Function"

_metals, doi:10.3390/met13101773_

Round 1

Reviewer 1 Report

This theoretical study shows an attempt to determine the activities of two-component melts based on partial functions using the symmetric and asymmetric methods. The radial distribution function polynomials have been applied to derive the pair potential energy of the liquid binary systems. The strengths of this work include the use of literature data of a large number of binary melts (36 binary systems) and four thermodynamic models to analyze the proposed methodology. The weakness of this work is the justification of the extension of the method, which is usually used for the gaseous state, to binary melts. I propose to discuss in more detail the validity of this approach.

However, there are a few issues that have to be revised or explained in the paper before publishing:

1) It is rather confusing that molecular distribution and intermolecular interactions were mentioned in the description of liquid metallic systems.  Providing a more accurate description (atomic distribution and interatomic interactions) of liquid binary systems in this paper may be more useful.  

2) The caption of Figure 6 should be corrected: it specifies Figure 5. It would be also more appropriate to specify systems in Figures 5 and 6.

3) Please check equation (1): the multiplier RT is missed.

It is a rather well-written paper. Only minor editing of English language is required.

Author Response

Dear Reviewer:
Thank you very much for your questions about my article, here are my answers:

1.I propose to discuss in more detail the validity of this approach.
I propose to discuss in more detail the validity of this approach. For the validity of this approach, I have added a very important model for liquid alloys - Miedema model and calculated SD and ARD for comparison, so that it is more convincing.
2. It is rather confusing that molecular distribution and intermolecular interactions were mentioned in the description of liquid metallic systems.  Providing a more accurate description (atomic distribution and interatomic interactions) of liquid binary systems in this paper may be more useful.
I have changed the description to atomic as you suggested.
3. The caption of Figure 6 should be corrected: it specifies Figure 5. It would be also more appropriate to specify systems in Figures 5 and 6.
I have corrected this typical error.
4. Please check equation (1): the multiplier RT is missed.
I have corrected this typical error.

Reviewer 2 Report

Strengths

Authors have study uses polynomials to describe the partial RDF in pure gases and extends this approach to binary liquid systems. The primary aim of this study is to characterize the molecular distribution and unravel intermolecular interactions, which are essential for accurate thermodynamic calculations. The RDF exhibits irregularities when the symmetric method is used instead of the asymmetric one for RDF calculation. Notably, the estimation of binary liquid alloy activity favors using the asymmetric method, especially when considering the average results obtained from both methods for the 36 binary liquid alloy systems investigated in this study. They selected and compared four thermodynamic models based on their symmetry degree. Data analysis reveals that the RSM exhibits the highest dependency on the symmetry degree. Conversely, the MIVM demonstrates superior adaptability to symmetric and asymmetric systems.

Weakness

1. Figure 2. (a) is illustration of the non-symmetric method of integration of radial distribution functions, (b) graphical representation of the symmetric method of integration of a radial distribution function, not Figure 2. an illustration of the non-symmetric method of integration of radial distribution functions, b graphical representation of the non-symmetric method of integration of a radial distribution function.

2. Where is Тable 1? And then the numbering of the Тables.

3. There is no reference to Figure 4 in the text.

4. Only one publication by the authors in the references. (13).

5. References should be updated.

Author Response

Dear Reviewer:
Thank you very much for your valuable comments on my article and the following answers to your questions:

1.Figure 2. (a) is illustration of the non-symmetric method of integration of radial distribution functions, (b) graphical representation of the symmetric method of 
integration of a radial distribution function, not Figure 2. an illustration of the nonsymmetric method of integration of radial distribution functions, b graphical 
representation of the non-symmetric method of integration of a radial distribution function. 

I will change the name of the picture as you requested.

2. Where is Тable 1? And then the numbering of the Тables.

Thank you for the valuable errors, I will check and correct them carefully.

3. There is no reference to Figure 4 in the text.

Thank you for the valuable errors, I will check and correct them carefully.

4. Only one publication by the authors in the references. (13).

For the literature, the derivation of this original model is cited, as well as its parameter representations, mainly citing the model.

5. References should be updated.

I will carefully check the content and revise the references.

Reviewer 3 Report

The manuscript presents the study of using the partial radial distribution function (RDF) to compute parameters of model pair potentials between binary liquid alloys using different thermodynamic models for the interaction. A symmetric method that uses symmetry of equal activities (and mole fractions) is used and compared to the presented asymmetric method that instead utilizes the pair RDFs. From what I understand, the symmetric method assumes symmetry in the pair RDF, although this was not clear for me in the manuscript (explained below). Four different thermodynamic models and their pair parameters are used to formulate the pair potentials between components. The asymmetric method removes the symmetric assumption and extends the distance range used from partial RDFs for the thermodynamic model pair potential calculations. The authors show the asymmetric method is more accurate than using the symmetric method and presents the thermodynamic models that have high dependence on symmetry versus adaptability between using either (less dependence on symmetry).

Overall, I think the manuscript is fairly well written but confusing in the structure and explanation for presenting the methods and results. The approach is valid and I think there is lots of value in this exhaustive comparison across so many binary pairs for alloys and the multiple thermodynamic models used. There are a few major and minor issues that must be clarified before considering the manuscript for publication. Attached is a list describing these issues along with some minor points aimed at improving the manuscript.

Author Response

Dear reviewer.
Thank you very much for your sound advice on my article, let me answer some of your questions below:
1.Maybe there is a better structure for the subsections?
Thank you very much for your valuable suggestions on the structure of my article, I will refer to your suggestions to modify the structure of the article.
2.Why for Figure5 is the SD all in one sub-plot(a) and ARD split up for(b-d)?And also, MIVM and RSM are in(b)but only Wilson in (c) and NRTL in (d)?
Also, there are two"Figure5"s(oneonpage11andoneonpage14).
I will modify the graph to put the Wilson and NRTL equations in one graph.
3.Where do the values in the text come from?
The symmetry in Section 4.1 is a measure of the symmetry of the standard activity of the binary liquid alloys and does not have any relation to the symmetry or asymmetry of the radial distribution function, whose data are the first 12 systems out of the 36 systems ranked according to symmetry from highest to lowest. The RSM model outperforms the remaining three models in six of these 12 systems, and the data are the average of the SD and ARD for these six systems.

Round 2

Reviewer 3 Report

Thank you for addressing most of my suggestions. I believe the manuscript is acceptable to be published in Metals.